# Intra-serotype variation of *Streptococcus pneumoniae* capsule and its quantification

Hannes Eichner,[1,2,3] Cindy Wu,[1] Michael Cammer,[4] Elizabeth N. H. Tran,[5] Timothy R. Hirst,[5] James C. Paton,[5,6,7] Jeffrey N. Weiser[1]

**ABSTRACT** *Streptococcus pneumoniae* (*Spn*) is a leading respiratory pathogen that depends on a thick layer of capsular polysaccharide (CPS) to evade immune clearance. Disease prevention by CPS-based vaccines is limited because of the species' high genome plasticity and ability to express over 100 different capsule types (serotypes). Generally, intra-serotype variations in capsulation are overlooked, despite the genetic variability of the bacterium. This oversight may result from a lack of standardized, reliable, and easily available methodology to quantify capsulation. Here, we have modified two methods to analyze the *Spn* capsule: immunoblot quantification of CPS in bacterial lysates and light microscopy to assess capsule thickness. Two assays were used because each measures distinct aspects of capsulation that could be differentially affected by the density of CPS. Quantification of either CPS amount or capsule thickness predicted the effectiveness of immune serum in opsonophagocytic killing assays for isogenic strains. Our standardized approaches both revealed significant differences in both CPS amount and capsule thickness among clinical isolates of the same serotype, challenging the assumption that intra-serotype capsulation is a conserved feature. As expected, these two methods show limited intra-strain correlation between amounts of CPS production and capsule thickness.

**IMPORTANCE** Despite the availability of vaccines, *Streptococcus pneumoniae* remains a leading cause of respiratory and invasive diseases. These vaccines target a polysaccharide capsule the bacterium uses to evade the immune system. Variation of the capsule composition subdivides the organism into serotypes and influences its protective potency. Another critical factor affecting this protection is capsule size. It is commonly assumed that *S. pneumoniae* strains of the same serotype produce capsules of consistent size, despite the organism's heterogeneity. In this study, we challenge this assumption by analyzing clinical isolates of the same serotype. Existing methods were modified to achieve high reproducibility and increase accessibility. Our data reveal significant fluctuations in capsule production within a given serotype. Our findings suggest that *S. pneumoniae* research should consider capsule size, not just its presence and type. The results imply that standardized vaccine efficacy tests may yield variable results depending on the capsule production of target strains.

**KEYWORDS** pneumococcus, immunoblotting, polysaccharides, image analysis, capsule, confocal microscopy, serotypes

The opportunistic pathogen *Streptococcus pneumoniae* (*Spn*, pneumococcus) commonly colonizes the human upper respiratory tract. Because spread to normally sterile sites results in disease, *Spn* is a leading cause of pneumonia, septicemia, and meningitis (1). This bacterium's virulence is largely attributed to its polysaccharide capsule, a complex structure that enables it to evade the host's immune system (2). To date, over 100 chemically and structurally distinct capsule types, or serotypes, have been

Address correspondence to Jeffrey N. Weiser, jeffrey.weiser@nyulangone.org, or Hannes Eichner, haneic.ac@gmail.com.

J.C.P. is a Director and Chief Scientific Officer of GPN Vaccines Ltd, T.R.H. is the Chief Executive Officer of GPN Vaccines Ltd, E.N.H.T. is a Research Scientist at GPN Vaccines Ltd. J.C.P. and T.R.H. hold an equity interest in GPN Vaccines Ltd. This does not alter adherence to policies on reporting and sharing of data, protocols, and materials.

See the funding table on p. 15.

*[This article was published on 14 February 2025 with an incorrect corrresponding email. The first author's email was corrected in the current version, posted on 24 February 2025.]*

identified (3, 4). Each has varying capabilities to protect the bacterium and contribute to disease (2). The widespread use of vaccines targeting the *S. pneumoniae* capsule has reduced the global burden of pneumococcal infection, but protection through immunization targets only ≤23 unique serotypes and thus remains incomplete (5, 6). Because vaccine-induced immunity is serotype dependent, there is a gradual selection for serotypes not included in current vaccines due to vaccine immune pressure (7, 8). This serotype replacement can be the result of not only increased fitness of non-vaccine serotypes but also the organism's remarkable genetic plasticity due to its natural transformability. In pneumococcal infection, successful lineages switch serotypes by exchanging the *cps* genetic locus (also called *cap* or *wzy*), which is responsible for the synthesis and surface expression of capsular polysaccharides (CPS) (9). While the 5′ section of the *cps* locus (*cps*A-D) is conserved, the diversity in the 3′ end enables the synthesis of the array of pneumococcal capsule types (10).

The capsule's size and composition play crucial roles in the pathogenicity of *Spn*, serving as a physical barrier against complement and antibody deposition, dehydration, and even as a carbon source during starvation (11–13). *S. pneumoniae* further exhibits a sophisticated regulation of CPS expression throughout its life cycle (10). Strains with a thicker capsule are more capable of evading opsonophagocytic clearance during invasive infection when the organism encounters higher levels of complement and antibody (14, 15). In contrast, adherence to host cells and mucosal colonization are enhanced with a thinner capsular coat (16–18). Many bacterial factors have been shown to affect the levels of CPS expression to allow for adaptation to different hosts and environments. These include numerous transcriptional and translational controls of the *cps* locus (19–23), as well as phase variation through recombination events within the *hsd* locus affecting global gene regulation (24–26).

Accurately measuring the capsule remains challenging due to a lack of highly sensitive, standardized methodologies for quantifying CPS production and/or capsule thickness. Previously described methods to quantify CPS depend on multiple antibodies to the same serotype, which are difficult to obtain (capture ELISA) (18), or semi-quantitative application of chemical dyes to visualize certain polysaccharides (27–29). Immunoblots are more accessible but not employed in a standardized fashion (15, 20, 30, 31). Fluorescent light microscopy can be used to create a negative stain of the capsule on the bacterium with limited resolution (12, 19). Electron microscopy can produce images with great detail, but true representative visualization can be questioned as sample processing may lead to distortion of the capsule (17, 32). As a result of this lack of accessible quantification and visualization, it is unclear how much capsulation varies in the highly heterogenous pneumococcal population. This inconvenient truth is largely overlooked despite the genetic plasticity of *S. pneumoniae*, the number of serotypes, and extensive regulation of the capsule. For example, surveillance studies often analyze the *cps* genetic locus but fail to investigate the expressed capsule itself (33). Vaccine studies often only consider the serotype when using standard tests for efficacy, but not intra-serotype variation of capsule thickness and effects on immune evasion (34).

Here, we describe and compare two tractable methods that measure two distinct aspects of the capsule: quantification of CPS produced by the cell or capsule thickness on the bacterial cell surface. The latter could potentially be affected by the density (or packing) of cell-associated CPS. We then show how these methods reveal the extent of intra-serotype diversity among clinical isolates.

## RESULTS

### Strains used to verify methods

A set of clinical isolates and mutants derived from these strains served as defined controls to validate our methods (Table 1). This included previously described serotype 6A isolates (P376 and P592) and spontaneous mutants (P385 and P2797), each with different single amino acid polymorphism in the initiating glycosyltransferase CpsE (15).

**TABLE 1** Bacterial strains

| Serotype | Number | Description | Reference |
|---|---|---|---|
| Reference strains | | | |
| 3 | P52 | Serotype 3, clinical isolate, collection of Robert Austrian | This study |
| 4 | P2406 | Serotype 4, streptomycin-resistant derivative of TIGR4 | (15) |
| 6A | P376 | Serotype 6A, clinical isolate, collection of Robert Austrian | (35) |
| 23F | P2499 | Serotype 23F, streptomycin-resistant derivative of P1121, a clinical isolate | (36) |
| Control strains | | | |
| 6A | P385 | Spontaneous variant of P376 that contains two missense mutations: CpsE$_{Phe297Ile}$ and YqfR$_{Val314Ile}$ | (37) |
| | P2752 | Corrected mutant of P385 by transforming P376 *cps* locus, restoring the *cpsE* mutation | (37) |
| | P592 | Clinical isolate from the collection of Robert Austrian | (37) |
| | P2797 | Mouse passaged variant of P592 containing missense mutation CpsE$_{Thr321Ala}$ | (37) |
| Other strains | | | |
| 23F | P426 | Serotype 23F, clinical isolate from the collection of Robert Austrian | This study |
| 23F | P2881 | Variant of P426 with increased colony opacity and size; contains missense mutation CpsE$_{Leu201Pro}$ | This study |
| 4 | P2422 | P2406 with the entire *cps* locus replaced with Sweet Janus cassette | (15) |
| 3 | P2090 | Serotype 3, American Type Culture Collection number 6303 | ATCC 6303 |
| Clinical isolates | | | |
| 3 | P2914 | Obtained from Adam Ratner at New York University Grossman School of Medicine, mastoiditis/meningitis patient | This study |
| | P2915 | Obtained from Anne Wyllie at Yale University, saliva | This study |
| | P2916 | Obtained from Anne Wyllie at Yale University, saliva | This study |
| | P2917 | Obtained from Anne Wyllie at Yale University, saliva | This study |
| | P2918 | Obtained from Anne Wyllie at Yale University, saliva | This study |
| | P2919 | Obtained from Anne Wyllie at Yale University, saliva | This study |
| | P2932 | Obtained from Anne Wyllie at Yale University, saliva | This study |
| | P2936 | Plate passaged P703, originally isolated as IAL-307 in 1996 | This study |
| | P2937 | Plate passaged P1584, obtained from C. Dowson at University of Warwick, blood isolate | This study |
| 4 | P15 | Obtained from Raymond Cornish, blood isolate | This study |
| | P30 | Blood isolate | This study |
| | P145 | Blood isolate causing hemorrhagic pneumococcal sepsis | This study |
| 6A | P302 | Clinical isolate from the collection of Robert Austrian | This study |
| | P306 | Clinical isolate from the collection of Robert Austrian | This study |
| | P307 | Clinical isolate from the collection of Robert Austrian | This study |
| | P448 | Clinical isolate from the collection of Robert Austrian | This study |
| | P461 | Clinical isolate from the collection of Robert Austrian | This study |
| | P503 | Clinical isolate from the collection of Robert Austrian | This study |
| | P1859 | Clinical isolate from South African Strain collection | This study |
| 23F | P5 | Clinical isolate, no information available except serotype | This study |
| | P19 | Clinical isolate from the collection of J. Glazer | This study |
| | P426 | Clinical isolate from the collection of Robert Austrian | This study |
| | P438 | Clinical isolate from the collection of Robert Austrian | This study |
| | P822 | Clinical isolate from the collection of Robert Austrian | This study |
| | P1091 | Clinical isolate from the collection of Nurith Porat | This study |
| | P1121 | Clinical isolate from the collection of T. Cates, nasal washes from colonized individuals inoculated with P833 | This study |
| | P1237 | Blood isolate, plate passaged from P26 | This study |
| | P1460 | Clinical isolate from the collection of T. Cates, plate passaged P833 | This study |
| | P1863 | Clinical isolate from South African Strain collection | This study |
| | P1900 | Clinical isolate from South African Strain collection | This study |

Also included was a corrected mutant of P385 that restored wild-type levels of CPS production (P2752).

## Immunoblots used for CPS quantification

The first method was based on the detection of CPS in lysates by immunoblotting using commercially available serotype-specific antisera (full protocol available in Supplemental material 1). Samples, equivalent by total protein measurement, were transferred to nitrocellulose membranes using a vacuum-coupled slot device (Fig. 1A; Fig. S1 and S2). A standard consisting of commercially available purified CPS was employed to generate a detectable range, according to which samples were loaded. An unencapsulated mutant (P2422) served as a negative control for non-specific binding. To allow comparisons across immunoblots, specific binding was expressed relative to a reference isolate of the same serotype. A complete representative immunoblot is shown in Fig. S1. Figure S2 shows that values obtained through our methodology are highly reproducible across independent experiments using lysates of the same strains. CPS was also quantified on culture supernatants but represented a marginal fraction compared to the cell pellet. Therefore, supernatants were not included in our analysis.

CPS quantification of the control strains confirmed that the CpsE point mutant of P376 (P385), which was no longer virulent in mice, expressed <10% of the CPS found in the parent (Fig. 1B) (15). As expected, full CPS expression was restored in the corrected mutant P2752. The CpsE point mutant of isolate P592 (P2797), selected for its ability to sustain invasive infection in mice, expressed >10-fold more CPS compared to its parent (15). The results of the control strains validated the sensitivity of our CPS quantification protocol.

## A modified dextran exclusion assay to measure capsule thickness

The second method quantified capsule thickness by light microscopy using a modified dextran exclusion assay (full protocol is available in Supplemental material 2). As in previous descriptions of dextran exclusion methods, total bacterial size, including the capsule, is detected as a shadow. Here, Nile Red staining instead of phase contrast microscopy was used as a simple and sensitive way to visualize the bacterial surface below the capsule (18, 38). Capsule thickness (or width) was then visualized in overlay images as the shadow from dextran exclusion extending over membrane lipids stained by Nile Red (Fig. 1C). Quantification of images was then performed *in silico*. We programmed a plugin for the Fiji package that analyzes changes in channel signal intensities across a line drawn perpendicular to the cell (detailed in Fig. S3). For each of the two channels, the positions where intensities reached 50% were recorded. Metadata from the image were then used to translate the positions of these intensity changes into the thickness of the capsule in micrometers. The limit of detection was 0.1 µm (represented by the unencapsulated strain P2422) (Fig. 1D). We measured approximately 40 cells of varying morphology (see Materials and Methods) per strain in three independent experiments. The mean of each experiment was analyzed as one data point. When tested on strains of four different serotypes, capsule width measurements with Nile Red staining were less variable and, in most cases, significantly greater compared with phase contrast microscopy (Fig. S4). This technique was also used to compare live and fixed *Spn* grown in liquid culture. No difference in capsule thickness was seen after the bacteria were fixed for 1–4 days (Fig. S5A). Interestingly, large variations were seen for the four serotypes tested when comparing the capsule width of *Spn* grown in liquid broth and agar plate (Fig. S5B).

For the control strains, the overall results of the capsule thickness quantification were consistent with those of the CPS immunoblot, although the magnitude of strain-to-strain differences varied between assays (Fig. 1D). P376 and P2752 capsules had similar thicknesses, whereas the P385 capsule was ~30% as thick. Measurements of P592 capsule thickness were below the detection limit, whereas the P2797 capsule thickness was comparable to P376.

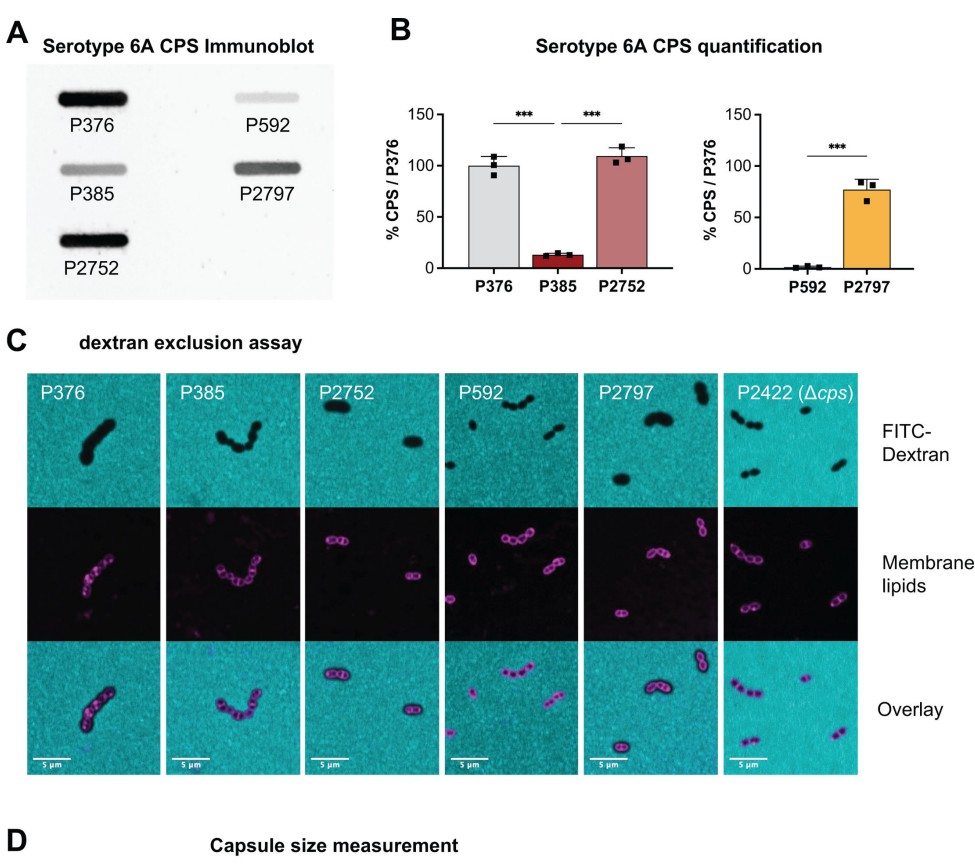

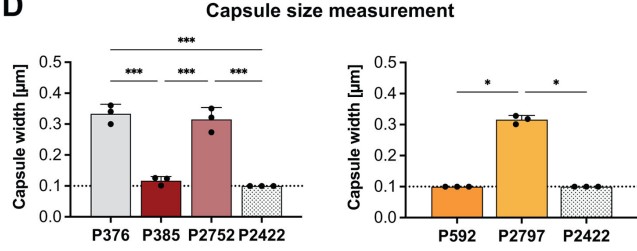

**FIG 1** Methods to analyze *S. pneumoniae* capsule. (A) Immunoblot of capsular polysaccharides of serotype 6A. P385 and P2752 are descendants of P376. P2797 is a descendant of P592. (B) Quantified CPS using densitometry on CPS immunoblots. Values were compared to the reference strain P376. (C) Capsule visualization via light microscopy using FITC-labeled dextran to visualize capsule as a shadow. Bacterial surface was visualized by staining membrane lipids with Nile Red. (D) Image analysis to measure capsule as the distance between bacterial surface and shadow edge in dextran exclusion assay. Dashed line represents limit of detection. (B and D) All data represent the mean of three independent experiments with standard deviation. Ordinary one-way ANOVA with multiple comparisons was performed for P376, P385, and P2752. Unpaired, two-tailed Student's *t*-test performed for P592 and P2797 in panel B. Kruskal-Wallis test comparing to P2797 performed for P592, P2797, and P2422 in panel D. *$P < 0.05$, **$P < 0.01$, and ***$P < 0.001$.

## Reproducibility of assays

We did not observe reproducibility problems between investigators in both methods. While the dextran exclusion assay was robust throughout, the immunoblot required strict adhesion to the preparation of the standard, reference strains, and samples. Unlike the dextran exclusion assay, the immunoblot is limited to relative quantification, which restricts its comparability between studies. On the other hand, the immunoblot assay is a measure of CPS volume and could be an inherently more sensitive measure of capsulation. Since the capsule forms a spherical layer around the cell, changes in thickness influence the overall volume in a nonlinear way. Variation in CPS may lead to only minor

thickness changes, which might not fully reflect the actual volume increase seen with the immunoblot.

## Electron microscopy confirms differences in capsulation

To further validate our two methods, the control strains were compared using transmission electron microscopy (TEM). We used a published protocol specifically modified to visualize the capsule of *S. pneumoniae* (Fig. 2) (32). The capsule is a strongly hydrated structure and water removal during TEM sample preparation may lead to its collapse. That effect may be prevented by adding lysine during the fixation process with glutaraldehyde (32, 39). As expected from the other assays, TEM showed that P376, P2752, and P2797 expressed a thick capsule layer on their surface. In contrast, P385 and P592 showed a scant amount of capsular material, which in the case of P592 did not appear to be sufficient to uniformly cover its surface.

## Variation among clinical isolates in capsulation

Our two methods were then used to assess the variability of capsulation in four sets of clinical isolates of different serotypes. Each set contained both carriage and invasive isolates obtained from multiple locations over many years. When compared to the reference strain P376, 8/8 serotype 6A isolates produced significantly different amounts of CPS (Fig. 3). For the capsule width determination, 4/8 of these isolates differed significantly from P376.

For a set of serotype 23F isolates, 5/11 and 9/11 differed significantly from the reference in CPS amount and capsule width, respectively (Fig. 4). Another serotype 23F strain (P2881), a spontaneous opaque colony variant of isolate P426, was also analyzed. As with the control strains, P2881 contains a single amino acid polymorphism in CpsE. The strain expressed >10-fold more CPS and had a capsule width >3-fold greater compared to its parent (Fig. S6). For serotype 6A and 23F strains in this study, sequence comparisons of the complete CpsE were provided (Fig. S7 and S8).

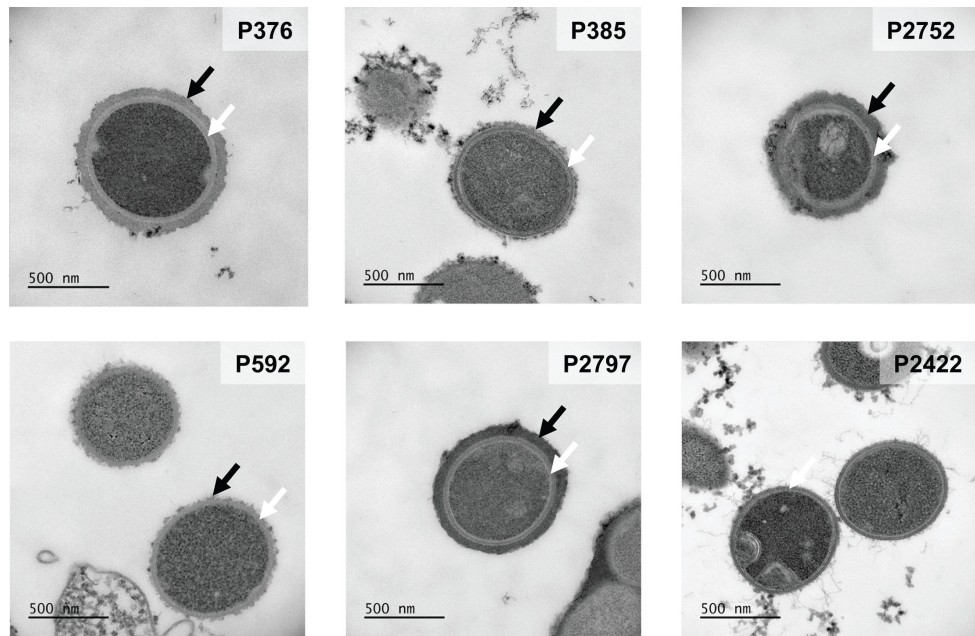

**FIG 2** Transmission electron microscopy to show the *S. pneumoniae* capsule. Bacteria were chemically fixed by including lysine acetate during the glutaraldehyde and formaldehyde exposure. P376, P385, P2751, P592, and P2797 are serotype 6A. P2422 is an unencapsulated (Δ*cps*) serotype 4 background. Samples were stained and postfixed using Ruthenium red and osmium and subsequently embedded in LR White. Black arrows mark the capsule, and white arrows mark the cell wall. Gamma adjusted for better contrast between the cell interior, cell wall, and capsule.

Because of its clinical importance in invasive infection, a set of serotype 3 isolates was also analyzed. Strains of this serotype form mucoid colonies due to increased levels of CPS and differences in its linkage to the cell surface. One out of nine and five out of nine isolates differed significantly in CPS amount and capsule width, respectively (Fig. 5). For this serotype, the capsule width was approximately five- to sixfold greater than the other serotypes tested.

Finally, a small set of serotype 4 isolates was tested because many pathogenesis studies use the TIGR4 strain (the parent of reference strain P2406). Two out of three and one out of three isolates differed significantly in CPS amount and capsule width, respectively (Fig. 6). Overall, our analyses of clinical isolates showed considerable strain-to-strain variability in both CPS production and capsule thickness. The findings also suggest that polymorphisms in CpsE could be a contributing factor to this variability among clinical isolates—a protein that has been shown to be a determining factor for capsule expression (15, 40).

## Greater capsulation increases immune evasion

Opsonophagocytic killing assays were carried out using pooled sera from rabbits pre- and post-vaccination with PCV13 or GPN Vaccines' candidate gamma-irradiated whole cell vaccine Gamma-PN3. The results were compared to the reference human serum

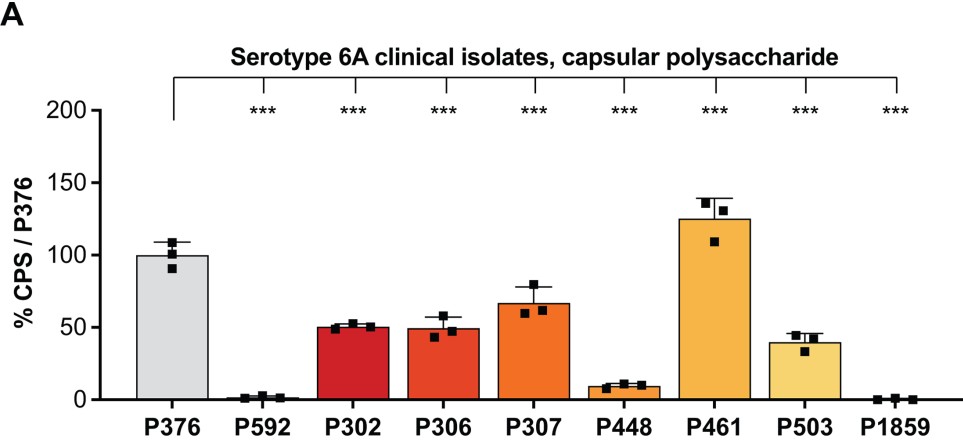

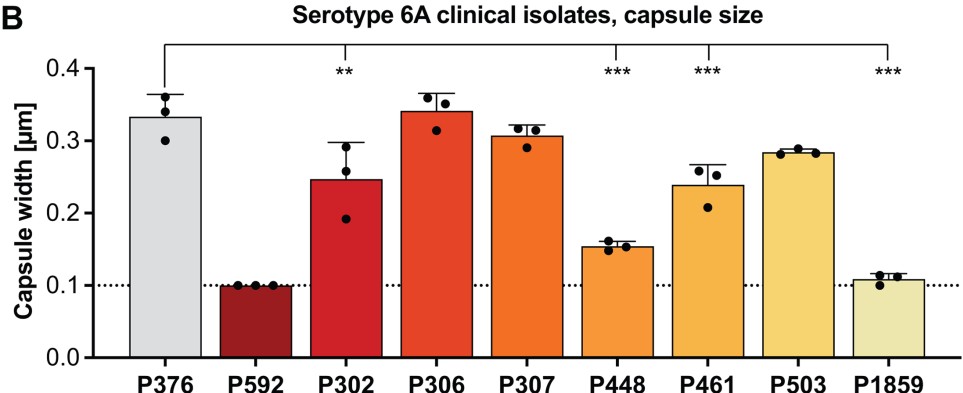

**FIG 3** Capsular polysaccharide production and capsule size of *S. pneumoniae* serotype 6A clinical isolates. (A) CPS production measured by immunoblot and densitometry shown relative to the reference strain P376. (B) Capsule thickness measured with our modified dextran exclusion assay. Dashed line represents limit of detection, based on the unencapsulated strain P2422. (A and B) All data represent the mean of three independent experiments with standard deviation. Data presented in panels A and B are from the same bacterial cultures. Ordinary one-way ANOVA was performed by comparing to P376. **$P < 0.01$ and ***$P < 0.001$.

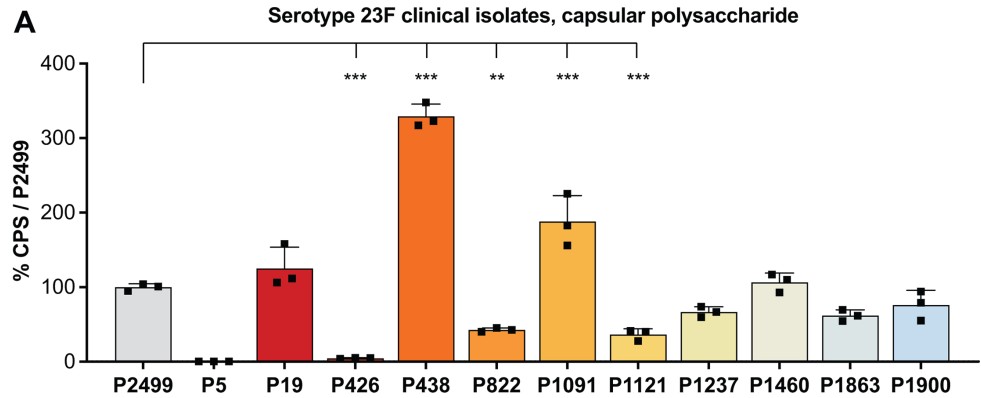

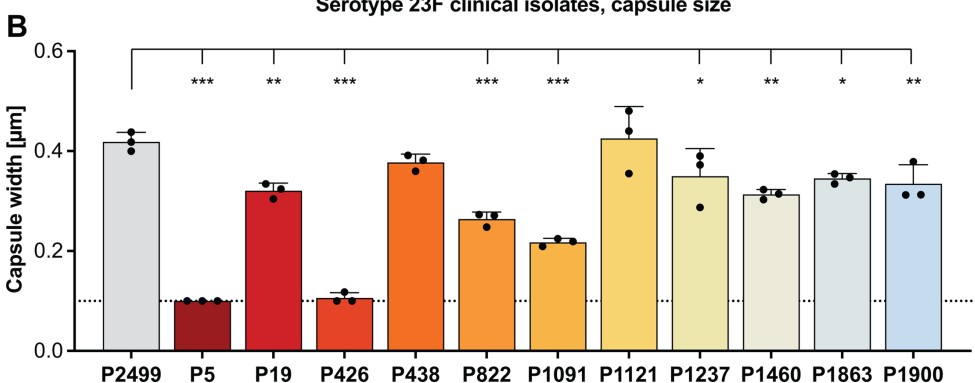

**FIG 4** Capsular polysaccharide production and capsule size of *S. pneumoniae* serotype 23F clinical isolates. (A) CPS production measured by immunoblot and densitometry shown relative to reference strain P2499. (B) Capsule thickness measured with our modified dextran exclusion assay. Dashed line represents limit of detection, based on the unencapsulated strain P2422. (A and B) All data represent the mean of three independent experiments with standard deviation. Data presented in panels A and B are from the same bacterial cultures. Ordinary one-way ANOVA was performed by comparing to P2499. *$P < 0.05$, **$P < 0.01$, and ***$P < 0.001$.

007sp. The assays compared the control strains to investigate the effect of variation of the *Spn* capsule in a standardized immune-mediated cell-killing assay used in vaccine efficacy assessments (Fig. 7). The poorly capsulated control strains P385 and P592 were observed to be highly sensitive to non-specific killing (NSK) by complement activity alone, showing 94% and 99% NSK, respectively (Fig. 7A) (41). Therefore, they did not meet the criteria of an NSK <70% suitable for further opsonic index (OI) titer analysis (41). Well-capsulated control strains, P376 and its corrected mutant P2752, showed similar OI titer values, whereas P2797 was more resistant (Fig. 7A). Killing was similar between P376 and P2752 when exposed to the 007sp standard or sera from rabbits vaccinated with Gamma-PN3 or PCV13 (Fig. 7C). As with the OI titers, P2797 was more resistant to killing than the other two highly capsulated strains (Fig. 7C and D). Together, these findings demonstrated the importance of variability in capsulation for widely used assays assessing correlates of vaccine protection.

## DISCUSSION

Existing methodologies for measuring bacterial capsules include chemical quantification of sugars, various immunoblot techniques, capture ELISAs, and microscopy. However, the absence of tractable and standardized protocols has hindered the comparability of results across different studies. Here, we try to bridge this gap by establishing reliable methods to measure CPS through immunoblotting and capsule surface expression via

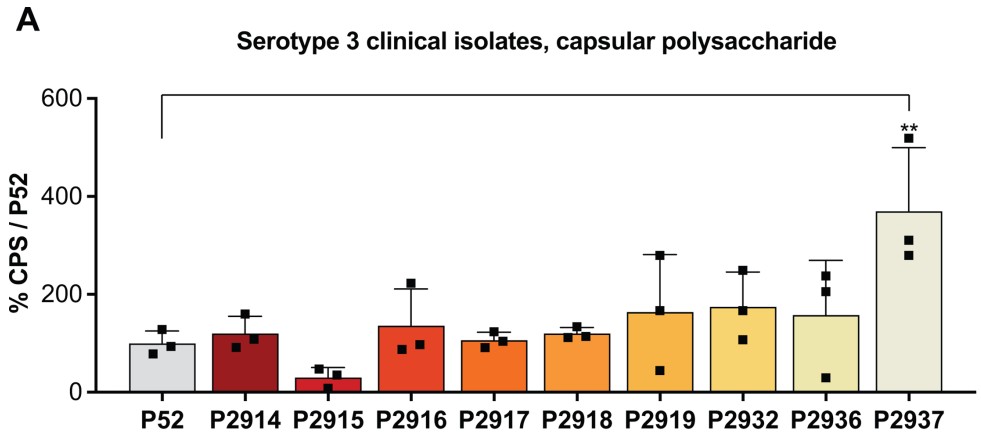

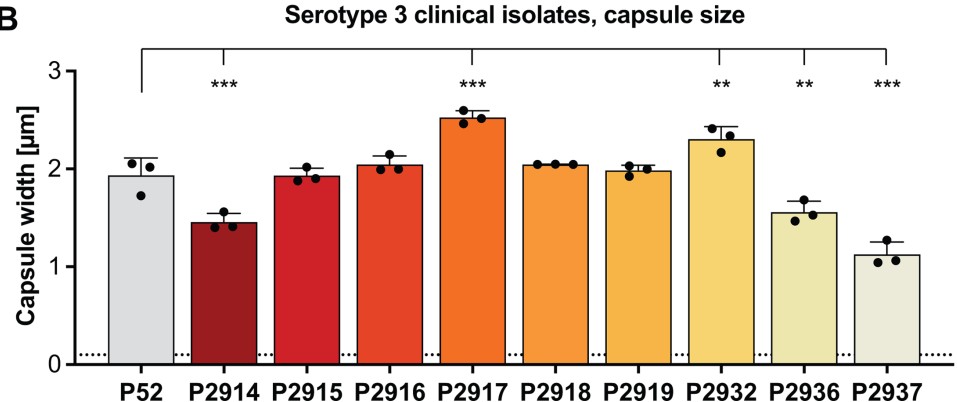

**FIG 5** Capsular polysaccharide production and capsule size of *S. pneumoniae* serotype 3 clinical isolates. (A) CPS production measured by immunoblot and densitometry shown relative to reference strain P52. (B) Capsule thickness measured with our modified dextran exclusion assay. Dashed line represents limit of detection, based on the unencapsulated strain P2422. (A and B) All data represent the mean of three independent experiments with standard deviation. Data presented in panels A and B are from the same bacterial cultures. Ordinary one-way ANOVA was performed by comparing to P52. $**P < 0.01$ and $***P < 0.001$.

microscopy. These methods were both used because each measures different aspects of capsulation that can be affected by the physical properties of CPS.

A challenge we faced in immunoblotting was the nonlinear acquisition of the electrochemiluminescence signal by cameras. These tend to automatically adjust to the brightest signal, which limits accurate quantification using a polysaccharide standard. This issue was addressed by loading sample volumes to fall into the linear section of the standard's signal. Additionally, variations in efficiency of antibody and development solutions necessitated the inclusion of a reference sample on each blot. This strategy allowed for robust relative quantification of CPS expression.

The dextran exclusion assay has long been used to characterize capsule thickness. Traditionally, this involves comparing bacterial surface in the transmission light field and dextran diffusion in the fluorescent light field. However, low contrast in the bright field requires visualization of the bacteria via phase contrast methods, limiting available focal planes and creating halo-like artifacts surrounding the cell. Thin capsules appear indistinguishable from those artifacts, skewing the use of this method to strains with thicker capsules. We addressed both issues by using the lipophilic and fluorescent Nile Red dye to visualize the bacterial membrane. Our approach allowed robust assessment of capsule width, including thinner types. It further works in several focal planes, greatly facilitating image acquisition and increased data point generation from one image. Additionally, the effectiveness of the Nile Red stain was dramatically enhanced by

## A

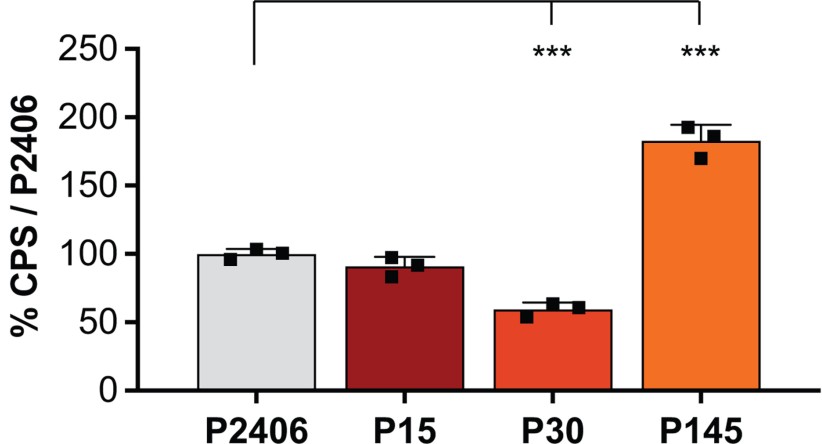

## B

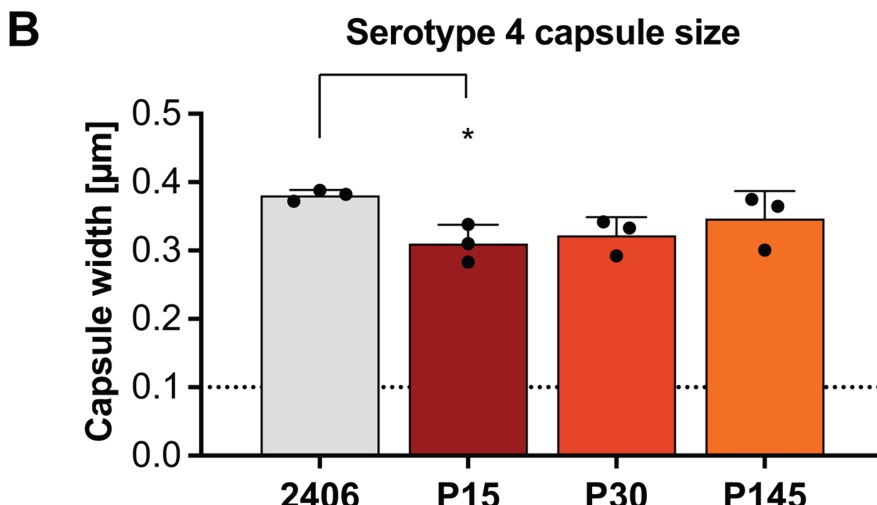

**FIG 6** Capsular polysaccharide production and capsule size of *S. pneumoniae* serotype 4 clinical isolates. (A) CPS production measured by immunoblot and densitometry shown relative to reference strain P2406. (B) Capsule thickness measured with our modified dextran exclusion assay. Dashed line represents limit of detection based on the unencapsulated strain P2422. (A and B) All data represent the mean of three independent experiments with standard deviation. Data presented are from the same bacterial cultures. Ordinary one-way ANOVA was performed by comparing to P2406. *$P < 0.05$, **$P < 0.01$, and ***$P < 0.001$.

immobilizing the bacteria with an agarose pad. The assay does not account for the cell wall outside the cell membrane, but there is no indication that its size fluctuates significantly (42). In accordance with that, no variation is visible between the P376 and P592 clinical isolates (Fig. 2).

Finally, we demonstrated that our modified dextran exclusion assay can be employed with live bacteria and cells that were chemically fixed with lysine and glutaraldehyde. Comparable results were achieved between live and fixed bacteria, suggesting the potential application of the method quantifying capsule size even on bacteria interacting with cells in culture or *in vivo*.

Analysis of the clinical samples revealed some discrepancies between the results of the two assays. For instance, while the control strain P2797 produced 75% of the

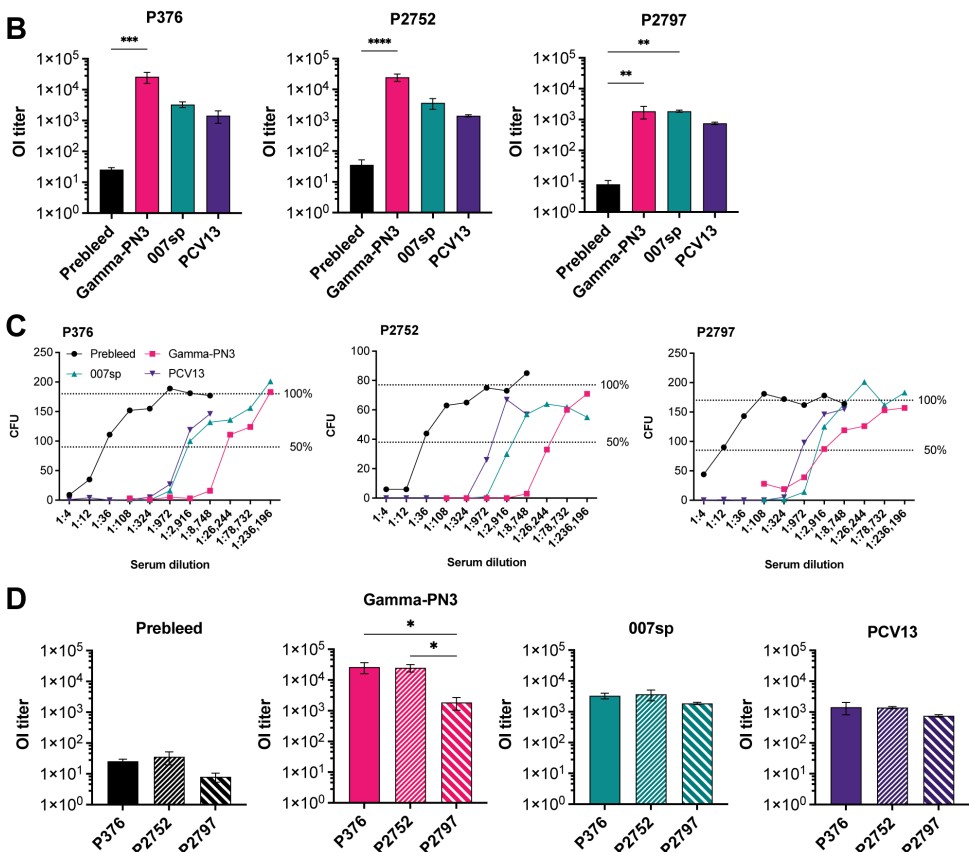

**FIG 7** Opsonophagocytic killing of serotype 6A control strains. (A) Table comparing the CFU counts in heat-inactivated complement (HI C′) and complement (C′) control wells (mean ± SD, $n = 4$), the non-specific killing percentages observed (mean ± SD, $n = 4$), and the mean opsonic index titers for each strain ($n = 3$, [#]$n = 2$). Strains were assayed in the presence of sera from pre-immunized rabbits (Prebleed), sera from rabbits immunized with Gamma-PN3 or with PCV13, and the human anti-pneumococcal reference serum 007sp (NIBSC); (**B**) OI titers for P376, P2752, and P2797 analyzed using one-way ANOVA followed by Dunnett's test with comparison to the Prebleed sample (mean ± SD). OI titers were not plotted for mutant strains P385 and P592 displaying >70% non-specific killing. (**C**) Representative killing curves observed for P376, P2752, and P2797. The 50% and 100% survival thresholds are indicated as dashed lines on each graph. (**D**) Statistical analysis on OI titers by serum vaccine groups. Data were analyzed using one-way ANOVA followed by Turkey's multiple comparisons test (mean ± SD). *$P < 0.05$, **$P < 0.01$, ***$P < 0.001$, and ****$P < 0.0001$.

CPS compared to P376, both strains had similar capsule sizes. Indeed, when correlating CPS production with capsule thickness in the four serotypes, we found a significant relationship only in serotype 6A isolates (Fig. S9).

These differences were not unexpected and are likely due to both approaches measuring different metrics, each with its own limitations. The immunoblot quantifies CPS without determining their location and cannot show their packing as a capsule. The dextran exclusion assay shows the surface expression of the CPS as a capsule but cannot visualize its density or differences in anchoring to peptidoglycan. Another reason for the discrepancy could be the dimensions measured. While the immunoblot analyses

the totality of CPS of the cell, i.e., a volume, the dextran exclusion assay only measures two dimensions. Therefore, changes in capsulation may have different impacts on the readout of the assays. Some of those discrepancies could be resolved using TEM. P592, for example, was found to express a capsular structure on its surface in insufficient amounts to fully cover the bacterium. It is not possible to say whether this observation is an artifact of sample preparation associated with TEM.

We demonstrated significant variability in capsule expression among isolates of the same serotype, a parameter largely ignored despite the diversity that characterizes the species. This variability underscores the need for comprehensive methods to quantify capsulation, particularly in studies comparing virulence and fitness.

Besides naturally occurring variation, other circumstances influencing capsulation should be considered during manipulations of *Spn* in the laboratory. A factor affected by CPS amount is genetic transformation—a thick capsule interferes with the DNA uptake during natural competence (43). Therefore, studies that rely on genetic manipulation may select for mutants with thinner capsules, leading to generally decreased fitness in invasive infection models. On the other hand, variants with thicker capsules will be selected during *in vivo* passage. Furthermore, it is standard practice in clinical and research laboratories to pick and expand single colonies for analyses. It is likely that the most prominent, largest, most mucoid colonies are typically selected from a diverse array of colony phenotypes and that this selection bias will tend to select for variants with higher CPS expression. Similarly, the impact of laboratory growth conditions on capsule production is often overlooked but may significantly influence downstream *in vitro* or *in vivo* experiments. Planktonic growth of various organisms in liquid culture may drastically alter gene expression when compared to solid-phase growth conditions (44–46). We demonstrated that the two different growth methods lead to an altered *Spn* surface capsule. These results underscore the importance of choosing the appropriate culture method with downstream experiments in mind, especially when virulence is assessed. The difference in surface capsule between planktonic and plate-grown bacteria was greatest for serotype 3 (Fig. S5A). Potentially, this derives from its capsule being secreted and not anchored to the peptidoglycan as with other serotypes, which may lead to less capsule being visible in the dextran exclusion assay. Accordingly, our results are only representative of the exact standard growth conditions studied here. Different environments may drastically affect capsule production, such as growth medium, pH, aeration, or temperature.

The methods presented here are tractable and shown to be highly sensitive for detecting the effects of minor genetic variations. This was demonstrated for spontaneous single amino acid changes of CpsE in the serotype 6A and 23F control strains. These were inducible by laboratory conditions as well as *in vivo* passage, indicating that such spontaneous changes could be a critical contributor to *Spn* pathogenicity. In another nasopharyngeal pathogen, *Neisseria meningitis*, immune killing pressure was demonstrated to quickly lead to capsule expression adaptation and was associated with invasive disease (47). Our methods could readily be applied to other encapsulated species.

Importantly, we show that capsulation has major effects in assays widely used as correlates of vaccine-mediated protection, such as opsonophagocytic killing assays (41, 48). However, we also show that strains of the same serotype and similar capsulation may show differing results in these assays. The effect can possibly be explained by the variation in the expression of pneumococcal proteins that impact complement deposition, e.g., CbpA and PspA (49, 50). Tests that investigate vaccine efficacy against only a single strain of *S. pneumoniae* may fail to take into consideration protection against a larger array of variants of the same serotype that differ in capsulation and other genetic factors.

## MATERIALS AND METHODS

### Bacterial strains and culture

*S. pneumoniae* strains used in this study are listed in Table 1 (15, 35–37). One strain was selected as a reference to compare values within a given serotype. Serotype 6A strains P376, P385, P2752, P592, and P2797 were used as control strains for establishing our methodology.

Plate cultures were grown on blood agar and tryptic soy agar plates. For liquid culture, serotype 6A and 3 strains were grown in Columbia broth, and serotype 4 and 23F strains were grown in tryptic soy broth. Bacteria were diluted 1:10 from a −80°C frozen stock prepared from a mid-log liquid culture. Optical density (OD) was measured using the Thermo Fisher Spectronic 200. After 2 h of growth, bacteria were diluted to an $OD_{620} = 0.15$ in a culture tube with a new medium. Samples were harvested at the mid-log phase. Although growth rates varied, we confirmed that cells were harvested in the exponential phase for all strains.

To prepare a frozen stock mid-log culture, bacteria were scraped from an overnight plate culture and suspended in the growth medium. After 2 h at 37°C, the bacterial culture was diluted to an $OD_{620} = 0.15$ in fresh medium and grown to $OD_{620} = 1.0$. One milliliter of bacterial culture was then pelleted, re-suspended in 300 µL of growth medium containing 20% (vol/vol) glycerol, and frozen at −80°C.

### Immunoblot

A detailed protocol can be found in supplemental material 1. A volume of 2 mL of bacterial culture for every strain of interest was centrifuged at $13,500 \times g$ for 1 min, and the pellet was lysed in 400 µL of 1% Triton-X 100 in 1× PBS. Due to the mucoid structure of serotype 3 strains, pellets were additionally washed with 1× PBS prior to lysis. Once lysis was complete, indicated by the solution turning from turbid to clear, 180 µL of the lysate for each strain was added to 30 µL aliquots of 3.33 µg/µL proteinase K, diluted in 1× PBS. These were incubated at 65°C for 15 min. Another 180 µL of the lysate for each strain was added to 30 µL aliquots of 1× cOmplete, EDTA-free Protease Inhibitor Cocktail (Roche, Ref# 11836170001), followed by an additional incubation at 37°C for 5 min. To standardize sample loading during transfer, the total protein concentration was determined by using the Pierce BCA Protein Assay Kit (Thermo Scientific). Loading of the immunoblot samples was standardized to a concentration of 500 ng of protein.

Sample transfer required a vacuum slot-blot device (Cole Palmer Minifold II Slot-Blot Manifold) to suction samples onto a nitrocellulose membrane (GE Healthcare Life Sciences, Amersham Protran 0.45 µm). Filter paper (GE Healthcare Life Sciences, Whatman Qualitative Filter Paper Grade 3) was dampened with 1× PBS. The membrane was then applied atop and aligned to the slots of the device before the top layer of acrylic was placed over the membrane. With the vacuum turned on, samples were applied to their designated slots to prevent diffusion into the membrane. Once the samples were suctioned through the slots of the device, each well was washed with 250 µL of 1× PBS. A 1:2 dilution series of purified capsular polysaccharides were transferred to the membranes as standard (Serotype 3 [Merck]: ATCC # 17-X; Serotype 4 [Merck]: ATCC # 173-X; Serotype 6A [Pfizer]: ATCC # 14-X; and Serotype 23F [Pfizer]: ATCC # 103-X).

After transfer, the membrane was first blocked with 3% skim milk in 0.1% PBS-Tween (PBST) for 30 min on a plate shaker. Then, the blot was washed with 1× PBS for 5 min. The membrane was then incubated with the appropriate primary antibody, serotype-specific typing antiserum (Statens Serum Institute Diagnostica, Serotype 3: Ref# 16746, Serotype 4: Ref# 16747, Serotype 6A: Ref# 16900, Serotype 23F: Ref# 16913; all: rabbit, final dilution 1:40,000 in 0.1% PBST) for 30 min with shaking. Because the serotyping antisera used are polyclonal, they were pre-treated with P2422, an unencapsulated strain, to absorb out unspecific binding to noncapsular antigens. Then, the blot was washed 2× for 10 min with 0.1% PBST. Finally, the membrane was incubated with the secondary

antibody, a goat-derived horseradish-peroxidase-conjugated anti-rabbit IgG (Invitrogen Ref# G21234, diluted 1:5,000 in 0.1% PBST), for 30 min with shaking. Between each step, the membranes were washed twice with 1× PBST for 10 min. All steps were performed on a plate shaker with gentle shaking settings.

The blot was developed with an electrochemiluminescence solution (Thermo Fisher, Super Signal West Femto Maximum Sensitivity Substrate) before being imaged with the iBright Imaging System (Invitrogen). Densitometry on the membrane image signals was performed using FIJI package (ImageJ version 2.14.0/1.54f, build c89e8500e4) (51).

## Dextran exclusion assay

A detailed protocol can be found in supplemental material 2. Fresh Staining Solution (2 mg/mL FITC-dextran [2,000 kDa] and 2.5 nM Nile Red) and an agarose-dye mix (1% [wt/vol] agarose in growth medium, 2 mg/mL FITC-dextran [2,000 kDa], and 2.5 nM Nile Red, kept liquid at 50°C–55°C) were prepared. Agarose pads can be created by adding 75 µL of the warm agarose-dye mix on a microscopy slide and placing an 18 × 18 cover slip (100 µL for 22 × 22) directly on top. Leaving the pads to dry for 5 min while the bacteria were harvested enabled easy removal of the cover slip by sliding it off. A volume of 1 mL of bacteria from a mid-log liquid culture was spun down at 10,000 $g$ for 1 min, the supernatant removed, and the pellet re-suspended in 100 µL Staining Solution. Bacteria were handled at room temperature as storage on ice seemed to trigger lysis in several strains. A volume of 5 µL of suspended bacteria in Staining Solution was placed on #1.5 (170 ± 5 µm thickness) borosilicate cover slips. The coverslips were placed on the agarose pads and sealed with nail polish. Prepared samples were successfully imaged for up to 3 h post-preparation. Images were acquired using the Leica Stellaris 8 Falcon laser scanning confocal microscope on an inverted DMi8 CS stand with Super Z Galvo and resonant scanner.

Image analysis was performed using a custom macro in Fiji (ImageJ Version 2.14.0/1.54f, build c89e8500e4) (51). Briefly, the macro creates a line of a given length consistently across all cells of interest, using image metainformation to translate pixels into micrometers. Along the line, the macro scans the intensity of the channel colors and creates an intensity peak. It further determines when the intensity curve reaches 50% intensity from the left and the right side of the curve. The distance between these points is translated into micrometers to measure the width of the bacterial cell (Nile Red channel) and the capsule shadow (dextran channel). The cell width is subtracted from the capsule shadow width and divided by two to calculate the capsule width on top of the bacterial surface. Cells for analysis were chosen to include cells in chains, diplococci, dividing cells, as well as large and small-sized cells. Measurements were also performed at the septum or cell equators.

## Transmission electron microscopy

Fixation and staining were carried out by the Electron Microscopy facility at NYU Grossman School of Medicine according to a published protocol (32). A volume of 10 mL bacterial culture from the mid-log phase was pelleted at 4,000 × $g$ for 5 min and the supernatant was removed. The pellet was re-suspended in 1 mL Lysine-ruthenium-red fixative solution (0.075% ruthenium red, 2% formaldehyde, 2.5% glutaraldehyde, and 1.6% [wt/vol] lysine acetate). The lysine must be added just prior to fixation. The fixation process was carried out on ice without extending over 20 min. Post-fixation and -staining were performed with osmium and 0.075% ruthenium red, respectively. The samples were embedded in LR White, and sections were stained with 4% aqueous uranyl acetate for 5 min. Images were acquired using the JEOL 1400 Flash transmission electron microscope with a Gatan 4k × 4k Rio CMOS camera and Gatan Elsa cryoholder.

## Opsonophagocytic killing assay

Single-plex opsonophagocytic killing assays were performed using the principles described by Nahm and Burton (41) for multiplexed opsonophagocytic killing assay

(41). Pooled sera were obtained from rabbits vaccinated with Gamma-PN3 or PCV13 (52). The human anti-pneumococcal capsule reference serum 007sp was purchased from the National Institute for Biological Standards and Control. Opsonic index (OI) titers were generated using the UAB's Opsotiter 3.0 MOPA analytical platform (41).

## ACKNOWLEDGMENTS

We thank Dr. Anne Wyllie of the Yale School of Public Health for providing clinical isolates. Additionally, we acknowledge the assistance of ChatGPT for its help with language refinement and input on code development for the Fiji macro used in this study. We acknowledge NYU Langone Health Microscopy Laboratory (RRID: SCR_017934) for providing microscopy services.

This work was supported by the NIH (5R01AI150893 and 5R37AI038446 to J.N.W.) and the Swedish Research Council (2021-06676 to H.E.). The microscopy shared resource is partially supported by Cancer Center Support Grant P30CA016087. The Leica SP5/8 confocal system is supported by the NIH (S10 RR024708).

## AUTHOR AFFILIATIONS

[1]Department of Microbiology, New York University Grossman School of Medicine, New York, New York, USA

[2]Department of Microbiology, Tumor and Cell Biology, Karolinska Institutet, Stockholm, Sweden

[3]Clinical Microbiology, Bioclinicum, Karolinska University Hospital, Stockholm, Sweden

[4]NYU Langone Health Microscopy Laboratory, NYU Langone Health, New York, New York, USA

[5]GPN Vaccines Ltd, Yarralumla, Australian Capital Territory, Australia

[6]Research Centre for Infectious Diseases (RCID), The University of Adelaide, Adelaide, South Australia, Australia

[7]Department of Molecular and Biomedical Sciences, The University of Adelaide, Adelaide, South Australia, Australia

## AUTHOR ORCIDs

Hannes Eichner http://orcid.org/0000-0002-1724-420X

Michael Cammer http://orcid.org/0000-0003-4930-1739

Elizabeth N. H. Tran http://orcid.org/0000-0003-1644-2287

James C. Paton http://orcid.org/0000-0001-9807-5278

Jeffrey N. Weiser http://orcid.org/0000-0001-7168-8090

## FUNDING

| Funder | Grant(s) | Author(s) |
| --- | --- | --- |
| HHS | National Institutes of Health (NIH) | 5R01AI150893, 5R37AI038446 | Jeffrey N. Weiser |
| Vetenskapsrådet (VR) | 2021-06676 | Hannes Eichner |

## AUTHOR CONTRIBUTIONS

Hannes Eichner, Conceptualization, Data curation, Formal analysis, Funding acquisition, Investigation, Methodology, Project administration, Software, Supervision, Validation, Visualization, Writing – original draft, Writing – review and editing | Cindy Wu, Conceptualization, Data curation, Formal analysis, Investigation, Methodology, Validation, Visualization, Writing – original draft, Writing – review and editing | Michael Cammer, Methodology, Resources, Software, Validation | Elizabeth N. H. Tran, Conceptualization, Data curation, Formal analysis, Investigation, Methodology, Validation, Visualization, Writing – review and editing | Timothy R. Hirst, Conceptualization, Formal analysis,

Resources, Supervision, Validation | James C. Paton, Conceptualization, Formal analysis, Resources, Supervision, Validation, Writing – review and editing | Jeffrey N. Weiser, Conceptualization, Data curation, Formal analysis, Funding acquisition, Investigation, Methodology, Project administration, Resources, Supervision, Validation, Writing – original draft, Writing – review and editing

## DATA AVAILABILITY

Data used to generate this study was uploaded to figshare. The repository contains raw files for microscopy images for the dextran exclusion assay, including processed images with Regions of Interest and associated measurements. Raw immunoblot images, including culture metrics and densitometry results, are available. Data manipulation files are deposited as GraphPad Prism files. The Fiji macro, CpsE alignments, protocols, and TEM pictures are also uploaded. The doi is https://doi.org/10.6084/m9.figshare.27633558.

## ADDITIONAL FILES

The following material is available online.

### Supplemental Material

**Supplemental material 1 (Spectrum03087-24-S0001.pdf).** Detailed protocol for CPS immunoblot.
**Supplemental material 2 (Spectrum03087-24-S0002.pdf).** Detailed protocol for dextran exclusion assay and Fiji macro.
**Supplemental material 3 (Spectrum03087-24-S0003.pdf).** Full blots from Fig S2A.
**Supplemental figures (Spectrum03087-24-S0004.pdf).** Fig. S1 to S9.
**Supplemental data (Spectrum03087-24-S0005.txt).** The code for the macro used to measure capsule thickness via imageJ.

### Open Peer Review

**PEER REVIEW HISTORY (review-history.pdf).** An accounting of the reviewer comments and feedback.

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
