## [Reviewer comments · Microbiology Spectrum]

Microbiology Spectrum

Intra-serotype variation of *Streptococcus pneumoniae* capsule and its quantification.

Hannes Eichner, Cindy Wu, Michael Cammer, Elizabeth Tran, Timothy Hirst, James Paton, and Jeffrey Weiser

Corresponding Author(s): Jeffrey Weiser, New York University

Review Timeline:

Submission Date:	December 9, 2024
Editorial Decision:	January 16, 2025
Revision Received:	January 16, 2025
Accepted:	January 20, 2025

Editor: Cezar Khursigara

Reviewer(s): The reviewers have opted to remain anonymous.

Transaction Report:

DOI: <https://doi.org/10.1128/spectrum.03087-24>

Re: Spectrum03087-24 (Intra-serotype variation of *Streptococcus pneumoniae* capsule and its quantification.)

Dear Prof. Jeffrey N. Weiser:

Thank you for the privilege of reviewing your work. Below you will find my comments, instructions from the Spectrum editorial office, and the reviewer comments.

I am pleased to inform you that your manuscript has been editorially accepted for publication. However, there are a few additional questions in the submission form that need to be answered before the final decision. Once these are completed, please return your submission so that I can move your paper forward to acceptance.

Revision Guidelines

Sincerely,
Cezar Khursigara
Editor
Microbiology Spectrum

Re: Spectrum03087-24R1 (Intra-serotype variation of *Streptococcus pneumoniae* capsule and its quantification.)

Dear Prof. Jeffrey N. Weiser:

Your manuscript has been accepted, and I am forwarding it to the ASM production staff for publication. Your paper will first be checked to make sure all elements meet the technical requirements. ASM staff will contact you if anything needs to be revised before copyediting and production can begin. Otherwise, you will be notified when your proofs are ready to be viewed.

Sincerely,
Cezar Khursigara
Editor
Microbiology Spectrum